# In Vivo Evaluation of PCL Vascular Grafts Implanted in Rat Abdominal Aorta

**DOI:** 10.3390/polym14163313

**Published:** 2022-08-15

**Authors:** Anna A. Dokuchaeva, Aleksandra B. Mochalova, Tatyana P. Timchenko, Kseniya S. Podolskaya, Oxana A. Pashkovskaya, Elena V. Karpova, Ilya A. Ivanov, Natalya A. Filatova, Irina Yu Zhuravleva

**Affiliations:** 1Institute of Experimental Biology and Medicine, E. Meshalkin National Medical Research Center of the RF Ministry of Health, 15 Rechkunovskaya St., 630055 Novosibirsk, Russia; 2Center of Spectral Investigations, Group of Optical Spectrometry, N.N. Vorozhtsov Novosibirsk Institute of Organic Chemistry SB RAS, 9 Lavrentiev Avenue, 630090 Novosibirsk, Russia; 3Departament of Physics, Novosibirsk State University, 2 Pirogova St., 630090 Novosibirsk, Russia

**Keywords:** tissue engineering, vascular scaffolds, in vivo rat model, electrospinning, polycaprolactone

## Abstract

Electrospun tissue-engineered grafts made of biodegradable materials have become a perspective search field in terms of vascular replacement, and more research is required to describe their in vivo transformation. This study aimed to give a detailed observation of hemodynamic and structural properties of electrospun, monolayered poly-ε-caprolactone (PCL) grafts in an in vivo experiment using a rat aorta replacement model at 10, 30, 60 and 90 implantation days. It was shown using ultrasound diagnostic and X-ray tomography that PCL grafts maintain patency throughout the entire follow-up period, without stenosis or thrombosis. Vascular compliance, assessed by the resistance index (RI), remains at the stable level from the 10th to the 90th day. A histological study using hematoxylin-eosin (H&E), von Kossa and Russell–Movat pentachrome staining demonstrated the dynamics of tissue response to the implant. By the 10th day, an endothelial monolayer was forming on the graft luminal surface, followed by the gradual growth and compaction of the neointima up to the 90th day. The intense inflammatory cellular reaction observed on the 10th day in the thickness of the scaffold was changed by the fibroblast and myofibroblast penetration by the 30th day. The cellularity maximum was reached on the 60th day, but by the 90th day the cellularity significantly (*p* = 0.02) decreased. From the 60th day, in some samples, the calcium phosphate depositions were revealed at the scaffold-neointima interface. Scanning electron microscopy showed that the scaffolds retained their fibrillar structure up to the 90th day. Thus, we have shown that the advantages of PCL scaffolds are excellent endothelialization and good surgical outcome. The disadvantages include their slow biodegradation, ineffective cellularization, and risks for mineralization and intimal hyperplasia.

## 1. Introduction

Currently, tissue-engineered vascular grafts (TEVGs) have become an object of wide scientific interest, especially in terms of vascular prosthetics. In particular, electrospun scaffolds made of bioresorbable polymers appear to be a perspective concept for vascular replacements, applicable for clinical use. These scaffolds are capable of transforming in situ into three-layer vessels, mimicking the autologous artery [1,2]. It is assumed that such a vascular device should be replaced by functioning autologous tissue within 1 year, otherwise it will undergo fibrosis and calcification [3]. This limitation is very important for choosing a polymer for the future artificial matrix. Furthermore, there are other significant requirements that reduce the use of well-known polymers and the demand for tailored synthesis of new compounds. Mechanical behavior, durability, extensibility, imitation of a native vessel in the replacement area, a lack of cytotoxicity in the polymer and the products of its degradation, a balance between the resorption rate and vascular wall formation de novo, and many other features are crucial for a good clinical outcome [4,5,6,7].

Undoubtedly, polymer properties should be evaluated in in vivo research, and it is necessary to account for how various species of laboratory animals may react differently to the same polymer, which is essential in terms of translational medicine [1,8,9,10].

All of the above is equally true for polycaprolactone (PCL), which has been used for TEVGs since the end of the twentieth century. It has been found that electrospun matrices made of pure PCL are highly rigid, have low compliance and degrade slowly. However, both the polymer itself and its degradation products are non-toxic and non-immunogenic [2,4,10,11,12]. In this regard, many authors have tried to improve the properties of electrospun PCL matrices by introducing different additives and combining them with other natural and synthetic polymers [2,4,5,12,13,14,15].

Yet, there is still not enough information about the in vivo transformation of vascular grafts, their scaffold fibers’ resorption and blood flow characteristics. A model of TEVG implantation in rat abdominal aorta is usually used for this purpose. Commonly, these studies include a postoperative follow-up, followed by histological and immunohistochemical research. Using a wide set of methods, such studies provide a characterization of the matrix, including its mechanical properties, step-by-step biodegradation and cellular content with immunohistochemical identification of endothelial cells. The literature data report that PCL grafts endothelialized well and showed no signs of inflammation at 90–180 days of observation, but the polymeric carcass was retained and endothelial hyperplasia appeared by the end of the follow-up period. Besides that, in describing cellular ingrowth, these studies report different dynamics [16,17,18,19].

Therefore, the objective of the present research was to evaluate the hemodynamic and structural properties of monolayered PCL grafts at 10, 30, 60 and 90 days after being implanted in a rat abdominal aorta model.

## 2. Materials and Methods

### 2.1. Polymer Composition

Pellets (3 mm in diameter) of (1.7)-polyoxepan-2-one (ε-polycaprolactone) with a molecular mass (Mn) of 80 kDa (cat. No 440744) obtained from Sigma-Aldrich (Sigma-Aldrich Co., St. Louis, MO, USA) were dissolved in pure chloroform (Vekton, Saint Petersburg, Russia) under constant shaking at 37 °C. Each solution was prepared separately on the date of scaffold manufacturing. In this study, a PCL solution with a density of 10% by mass was used.

### 2.2. Vascular Scaffold Fabrication

Tubular scaffolds were manufactured at the environmental temperature of 37 °C and humidity of 22% in a NANON 01-B electrospinning setup (MECC Inc., Fukudo Ogori-shi, Japan) using a standard clip spinneret and a 2 mm rod rotary collector. The technical parameters were set as follows: applied voltage of 16 kV, feed rate of 0.5 mL/h, collector rotation speed of 300 rpm, spinneret speed of 150 rpm, tip-to-collector distance of 15 cm, needle diameter of 27G, nozzle cleaning interval of 59 s, and solution volume of 0.125 mL. We previously proved that the combination of these parameters provides the best mechanical properties and optimum pore size [20].

### 2.3. Scanning Electronic Microscopy (SEM)

SEM images of scaffold and dried explanted graft samples were obtained using a SU1000 FlexSEM II scanning electron microscope (Hitachi, Tokyo, Japan). Specimens were appropriately fixed on a specimen stub with conductive tape. Sample observation was performed using a backscattered electron detector at an electron beam energy of 15 keV and a pressure of 30 Pa. Every specimen was examined at 100×, 250×, 450×, 700×, 800×, and 1000× magnification; then, ten 700× fields of view were selected at each sample for the fiber size measurements. Linear pore size measurements were performed in ten images obtained at 700× magnification. For this, 20–40 measurements of the distance between the fibers laying in the surface plane were performed on each of the images. The pore size was calculated as the mean of the obtained data. All of the measurements were carried out automatically using the FlexSEM 1000 operating program.

### 2.4. Graft Sterilization

Prepared matrices were sterilized with ethylenoxide (750 mg/L) in a Steri-Vac 5XL sterilizer (3M, St. Paul, MN, USA) at a chamber temperature of 37 °C, air humidity of 70% and a sterilization cycle time of 3 h. Aeration was performed at the same temperature and took 8 h. Sterilization at 37 °C does not affect the mechanical properties, pore size and fibers’ ultrastructure of the scaffolds (S1).

### 2.5. In Vivo Implantation

All experimental procedures were carried out in accordance with EU Directive 2010/63/EU for animal experiments and were approved by the Ethics Committee of E. Meshalkin National Medical Research Center (protocol No 3, 15 June 2021).

Randomly, 10-month-old male Wistar rats (*n* = 40) with a 450–500 g body weight were divided into four experimental groups (10, 30, 60 and 90 days of implantation, respectively). On the day of operation, the animals were placed in individual cages with free access to drinking water and were deprived of food.

#### 2.5.1. Premedication

Premedication included sedation with dexmedetomidine (0.1 mg/kg) (Dexdomitor, Orion Inc., Espoo, Finland) via intramuscular injection in a hind limb. If not effective, the dosage was gradually increased up to 0.3 mg/kg. In the other hind limb, carprofen (5 mg/kg) (Rycarfa, KRKA d.d., Novo Mesto, Slovenia) was injected for pain relief. Atropine sulfate (0.05 mg/kg) (Dalhimpharm, Khabarovsk, Russia) was administered subcutaneously. A lateral tail vein was catheterized with a 24G plastic catheter (KDM, Berlin, Germany).

#### 2.5.2. Preoperational Procedures

After the premedication, the abdominal fur was shaved off with a veterinary trimmer (Moser, Unterkirnach, Germany), then the animal was fixed on the operation table laying on its back. Sevoflurane (3%) (Medisorb, Perm, Russia) was administered via the nasal mask of a gas-flow anesthesia system for rodents (Ugo Basile, Gemonio, Italy) under an air flow of 0.5 l/min. The working field was swabbed with 10% povidone-iodine solution (Kormend, Matyas kiraly ut 65, Hungary) and 95% alcohol (Kemerovo Pharmaceutical Factory, Kemerovo, Russia).

#### 2.5.3. Surgery

When the surgical stage of narcosis was reached, basic midline laparotomy was performed. First, 0.5% lidocaine (1 mL) (Dalhimpharm, Khabarovsk, Russia) was poured intraperitoneally. If muscular contractions of the abdomen occurred, the dosage of lidocaine was increased up to 3 mL as necessary. The intestine was moved out, washed with 9% NaCl solution (Solopharm, St. Petersburg, Russia) and wrapped in a sterile wet gauze. The abdominal aorta was dissected from the renal arteries down to the bifurcation. At this time point, Fraxiparine (25 IU) (Aspen Notre-Dame-de-Bondeville, Notre-Dame-de-Bondeville, France) was injected subcutaneously. After a two-minute pause, full aorta occlusion was performed: the upper clip was placed past the renal arteries, and the lower clip was placed above the bifurcation. A 5 mm section of the abdominal aorta was excised and an end-to-end anastomosis between the aorta’s upper section and a PCL graft (5 mm length × 2 mm in inner diameter) was performed. The same anastomosis was conducted on the lower section; both used an Optilene 8/0 thread (Braun, Rubi, Spain) (Appendix A). Cefazoline (1 mg/kg) (Biosintez, Penza, Russia) was administered in solution intraperitoneally. The operation wound was closed with a monofilament Optilene 5/0 thread (Braun, Rubi, Spain), and the cutaneous suture was swabbed with povidone-iodine solution.

#### 2.5.4. Postoperative Period

During the postoperative period, all of the animals were housed in individual cages with free access to drinking water and food. Postoperative treatment included Fraxiparine (15 IU) subcutaneously for three days after the operation, and carprofen (5 mg/kg) subcutaneously for two days after the operation.

During the whole follow-up period, the surgical outcome was evaluated daily, with specific attention given to the overall daily activity, movement in the hind limbs, defecation and urination. Euthanasia was preceded by an angiography and ultrasound diagnostic.

### 2.6. Angiography

Before the procedure, all of the animals were sedated with dexmetamidine and catheterized with a 24G catheter as described above.

CT scans were obtained using a small animal radiation therapy platform (SmART+, Precision X-ray Inc., North Branford, CT) that combines a high-accuracy cone-beam CT (CBCT) imaging system and a high-dose-delivery, therapeutic X-ray source built into a single platform. The radiation source is an X-ray tube with dual focal spot sizes and is used for both imaging (60–80 kVp, small focus with 1 mm Al filtration) and therapy (225 kVp, large focus with 0.15 mm Cu filtration). The imaging system uses a flat-panel, amorphous, 20 cm × 20 cm (1024 × 1024 pixel) silicon detector set in the opposite position of the X-ray source. For each animal, two series of CBCT scans were performed. The first series without contrast was carried out to image the animal anatomy and set the scanning boundaries. To enhance vessel visualization, the Ultravist 370 contrast medium (732 mg/kg) (Polysan, St. Petersburg, Russia) was injected continuously via the tail vein prior to (15 s) and during (10 s) CT imaging that continued for 50 s.

### 2.7. Ultrasound Diagnostic

Before the procedure, all of the animals were sedated with dexmetamidine, the abdominal area was shaved with a veterinary trimmer and then the Mediagel ultrasound gel (Geltek, Moscow, Russia) was applied. The ultrasound procedures and Doppler measurements were carried out using a Philips CX50 portable vascular ultrasound machine (Philips, Bothell, WA, USA) with a linear transducer. Peak systolic (PSV), end diastolic (EDV) and time average (TAV) blood flow velocities were measured at three topographic points such as in the abdominal aorta above the proximal anastomosis, and at the midline of the graft and abdominal aorta between the distal anastomosis and bifurcation. The resistance index (RI) was calculated for each animal as:$$\;\mathrm{RI}=\frac{\left(\mathrm{PSV}\;-\;\mathrm{EDV}\right)}{\mathrm{PSV}}$$

### 2.8. Euthanasia and Autopsy

By the end of the follow-up period (10, 30, 60 and 90 days) the animals were euthanized with an overdose of sevoflurane. The abdominal aorta fragments containing the graft were dissected and excised via abdominal access and rinsed in 0.9% NaCl; excessive moisture was removed with a gauze.

### 2.9. Histology

All of the samples were fixed in 10% buffered formaldehyde solution (Biovitrum, St. Petersburg, Russia) for 48 h, then samples were paraffinized in an automatic histological processor (Slee medical, Mainz, Germany) with Surgipath Paraplast X-tra paraffin (Leica, Germany) to make paraffin tissue blocks in a paraffin-embedding station (Microm, Waldorf, Germany). Using a rotary microtome HM 340 E (Microm, Waldorf, Germany), 6 µm sections were made. Obtained sections were stained with hematoxylin and eosin (Biovitrum, St. Petersburg, Russia) following a standard protocol; for specific staining, a Russell–Movat pentachrome staining kit (Diapath, Martinengo, Italy) and von Kossa staining kit (Biovitrum, St. Petersburg, Russia) were used according to the manufacturer’s instructions. Light microscopy was carried out at ×50, ×100 and ×400 magnifications using an Olympus CX31 (Olympus, Tokyo, Japan) laboratory microscope with LCmicro 2.2 image analysis software (Olympus Soft Imaging Solutions GmbH, Münster, Germany). For cellularity evaluation, ten 0.01721 mm2 fields (×400 magnification) in the scaffold area only were selected from H&E-stained slides of each sample. The cells were counted under visual control. The eventual data were represented as a mean and standard deviation for the cell count per observation field for each experimental group.

### 2.10. Statistical Analysis

Statistical analysis was performed using the Statistica 8 software (TIBCO Software Inc., Palo Alto, CA, USA). The results were shown as a mean and standard deviation (σ). The non-parametric Mann–Whitney U test was used as a statistical solution with a statistical significance set at *p* < 0.05.

## 3. Results

### 3.1. Non-Implanted Scaffolds

According to SEM measurements, the obtained scaffolds had an average fiber diameter of 4.776 ± 0.546 µm, average pore size of 14.994 ± 3.253 µm and average graft wall thickness of 54.1 ± 7.374 µm, which demonstrated the good replicability of PCL matrices whose fibers were arranged in a dense looped pattern. The overall scaffold structure before implantation was even and supported the original cylindrical shape (Figure 1).

### 3.2. PCL Graft Observation in Dynamics

After blood flow restoration, none of the implanted scaffolds showed signs of rupture or leakage (Figure 2).

All the animals were active on the first day of the postoperative period. No death or circulatory disorders were registered.

In spite of several cases of graft/aorta diameter mismatch when the graft diameter was larger than the vessel in a section point, no leakage, aneurisms, suture ruptures or thromboses were found on ultrasound and CT images (Figure 3).

The obtained ultrasound examination data revealed the main patterns in the blood flow behavior: the peak systolic and time average blood flow velocities tended to decrease along the way from the proximal aorta to the bifurcation, and likewise, the end diastolic flow velocity tended to increase, which resembled that of a native vessel and confirmed no stenosis in both proximal and distal anastomoses. A short tendency for all the velocity values to increase in dynamics was observed, but the large individual variability of these indicators made it difficult to interpret this trend clearly (Figure 4). However, the main indicator reflecting vascular resistance and compliance (RI) was stable during the entire follow-up (Figure 4), which was also due to the absence of stenosis and deformations in the implantation zone. Therefore, all 40 implanted prostheses were permeable, and no aneurism or stenosis was found.

### 3.3. PCL Graft Transformation in Dynamics

Histological analysis showed that ten days past the implantation, a moderate foreign body reaction persisted and first signs of endothelialization appeared (Figure 5).

The luminal surface of these prostheses was devoid of thrombosis or signs of fibrin; the dark red stain on the Russell–Movat pentachrome slides revealed a forming endotheliocyte monolayer to cover most of the graft. No signs of intimal hyperplasia were found at this time point. The thickness of the PCL matrix was infiltrated with lymphocytes, macrophages and giant cells; very few fibroblasts could be found. The pale yellow staining marked the presence of collagen in the forming adventitia.

The active remodeling process begun on the 30th day. Although the inflammatory response manifesting in moderate lymphocyte and macrophage infiltration remained, neoangiogenesis and an increasing migration of fibroblasts in the PCL matrix core were noted. Since that time point, giant cells tended to collect closer to the graft’s adventitial surface. The luminal surface was covered by a developed neointimal layer with signs of regionary hyperplasia that originated from the myofibroblasts laying underneath the endothelial layout. At that time, no metaplastic changes were found (Figure 5).

The most stable scaffold ingrowth with high cellularity was registered at 60 days after implantation (Figure 5 and Figure 6).

Giant cell infiltration was found on the margin between the graft and the forming neoadventitia surrounding the polymer. The scaffolds’ thickness was mostly filled with fibroblasts, and myofibroblasts also tended to gather on the outer half of the scaffold. The extent of intimal hyperplasia also significantly increased, as most of the samples revealed an enlargement of the subendothelial layer and chondroid metaplasia (Appendix A) of the cells within. At the same time, on the 60th day of observation the first signs of prosthesis calcification were found on the H&E slides, which was confirmed by SEM element analysis (Appendix A) that showed high concentrations of calcium and phosphorus in these formations, indicating calcium phosphate crystals (Figure 7). Another finding was the presence of silicium in these clusters. Points of calcification related to the sites of chondroid metaplasia were located subintimally, on the border of a PCL matrix.

By the 90th day after implantation, the graft cellularity greatly decreased (*p*_60–90_ = 0.02), whereas the cell infiltration in these samples was mostly represented by the rare inflammatory cells surrounding the scaffold from the side of the neoadventitia, and the fibroblasts migrating to the outer layer of the graft. The adventitial edge of the graft was also infiltrated with giant cells. The midparts of these matrices appeared to have poor cellular content, represented mostly by the single fibroblasts located far apart from one another. Most samples showed endothelial hyperplasia and several cases of calcification, which was also confirmed by SEM element analysis. These samples demonstrated poor inner vascularization, and only a few capillaries were found in the prepared sections.

The Russell–Movat pentachrome staining also revealed cellular infiltration and myofibroblastic component migration from the matrix’s middle to its adventitial margin through the time points, which led to the gradual development of neoadventitia. Most of the intensive collagen-specific staining matched the peak of graft ingrowth, indicating the formation of structured neoadventitia at the 60th day. No featured collagen-specific staining occurred within the scaffold (Figure 5), which was consistent with the cellular outflow that occurred at the 90th day of implantation. No sign of thrombosis was found by any of the listed methods. With time, the neointimal layer demonstrated less muscle- and nuclei-specific staining and became bright blue, which was seen on the microphotographs at 30–90 days of implantation and indicated the presence of proteoglycans.

SEM of dried explanted samples after 90 days of implantation demonstrates the thinning of polymer fibers to 2.013 ± 0.504 µm, which confirmed our suggestion concerning the slow rate of biodegradation of this polymer in vivo (Figure 8). However, no ruptures were found, and the graft retained its initial fiber architecture. SEM also confirmed cell permeability.

While comparing the light microscopy images of paraffinized samples and SEM images of dried samples of the same explanted material, vast differences between the microarchitecture and structure of nanofibers was revealed (Figure 9). The structure on the SEM images of dried samples remained close to the original: pores of polygonal shapes were formed by discernable fibers, laying in several planes; thus, the matrix retained its multiplanar architecture and fiber-to-fiber connections were visualized well. In paraffinized specimens, separate fibers were indistinguishable, and pores gained apparent margins of circular shape with a smooth transition from one plane of the section to another because of the polymer melting.

Noting the alterations in the scaffold structure after the paraffinization procedure, all of the measurements and conclusions concerning fiber and pore size and organization were based on the SEM images.

## 4. Discussion

In our previous work, we had shown the way the structure and mechanical properties of PCL matrices are influenced by the changes in manufacturing modes of an electrospinning setup [20]. Choosing a manufacturing mode for the scaffolds in this study, we were guided firstly by such a requirement: the graft strength should be close to that of the rat aorta. This allows us to achieve anatomical compatibility between the graft and the recipient vessel to ease anastomosis formation. Postoperatively, this approach decreased the risk of stenosis. Values of abdominal aortic wall thickness greatly vary among research groups [21,22,23]. We relied on the values (50.4 ± 3.3 µm) given by Y. Bezie et al. [24], and on our own measurements. Therefore, a thickness of 54.1 µm was chosen as optimal among those provided, both for better in vivo performance and for the convenience of surgical procedure. Unquestionably, the graft thickness not only affected the mechanical properties, but also the pore size [20,23].

Previous studies had shown that a pore size of ~30 µm was optimal for small arterial grafts as it provided better conditions for tissue formation [25,26,27,28]. Despite the smaller size (~15 µm), pores of our graft were permeable for cells that take part in tissue remodeling, as histological studies show. In the meantime, the histological picture indicates increasing cellular ingrowth up to 60 days after implantation and a decrease in cellularity by the 90th day of implantation (Figure 5 and Figure 6) due to the reasons not associated with the matrix, as it did not lose its architecture and the pores were permeable for cells. The reason for this may lay with the products of the polymer degradation that accumulated within the thickness of a graft, as other conditions remained the same within the observation period.

Calcification of vascular and valve grafts was identified to be one of the main problems of contemporary vascular prosthesis engineering [9,29]. Several materials, including biodegradable polymers [30] that were offered for clinical trials, have a significant risk of calcification after implantation, which makes it necessary to evaluate their calcium-binding capacity in vivo at long-term time points. In our case, mineralization was related to intimal hyperplasia and metaplastic processes, taking place on the margin between the grafts and growing neointima. Some authors [18,19] report similar data and mention transmural PCL graft calcification and tissue ossification in a few months after implantation. These studies suggest that chondroid metaplasia of the smooth muscle inlayer is involved in vascular mineralization in vivo, but the mechanisms of this process have not been studied yet.

Endothelialization is an obligatory requirement for a vascular graft, as the luminal surface stays in direct contact with the blood stream; any other type of surface exposes the graft to a higher risk of thrombosis. Simultaneously, intimal hyperplasia is a significant problem for PCL matrices. In this study, intimal hyperplasia was not critical for the graft patency, but it plays a role in graft calcification that may cause further ruptures, aneurisms, anastomosis leisure and, perhaps, detachment of the hyperplastic layer. As mentioned above, at the 60th day after implantation, samples showed a congruent endothelial layer spread upon most of the graft with featured signs of intimal hyperplasia.

Giant cell agglomerates on the adventitial side of the scaffold can be regarded as a chronic reaction to a foreign body, which remained active on the 90th day after implantation. These cells are functionally dedicated to phagocytosis and are often revealed in the histological picture of a granuloma inflammatory process [31], which is supposed to cause sclerosis due to phagocytic monokines. In combination with tissue calcification, further transformations of PCL grafts appear as fibrous encapsulation from the adventitial side and mineralization from the intimal margin.

An interesting SEM finding is the striking difference between the genuine scaffold architecture and the samples undergoing the paraffinization procedure. All the explanted samples that underwent conventional histoprocessing protocol were noted to show severe alterations in the scaffold structure as soon as 10 days after implantation, which goes against the known data [16,17,18,19,32]. Then, several sections taken from the 90-day group were dried and studied by SEM while the other parts underwent paraffinization in a histological processor. A comparison of the SEM images of paraffinized and dried fragments obviously revealed the big difference in structure between these approaches to tissue processing (Figure 9). This made us take a fresh look at the approaches to studying scaffold transformation.

As the melting point of PCL is 59–64 °C, depending on the chain length and molecular weight, high-temperature paraffin embedding is not suitable for this material as paraffin has a similar melting point. The conventional paraffinization procedure is recommended to be performed at 60 °C [33]. However, this temperature may deform or melt polymer fibers. Yet, SEM cannot give researchers a full picture of graft cellularization; therefore, we used a common histological analysis for cell content and ingrowth evaluation, but SEM is irreplaceable for fiber condition evaluation.

In this regard, it is also necessary to mention the difficulties with regard to sterilization of the PCL scaffolds, which was due to their melting temperature. We suppose ethyleneoxide sterilization was only possible using a “cold” (37 °C) cycle (Appendix A).

Although polycaprolactone is a well-studied polymer that is currently used for medical applications, the data concerning its in vivo transformations in animal models are very contradictory and require a more detailed investigation.

## 5. Conclusions

In this study, we showed such advantages of PCL vascular grafts as excellent endothelialization and good surgical outcome, along with their major limitations of slow biodegradation, ineffective cellularization and risks of mineralization and intimal hyperplasia. The mineralization of these grafts takes a specific pathway and is possibly related to intimal hyperplasia and chondroid metaplasia of the cells, forming the deeper layers of neointima. At the same time, these scaffolds demonstrated good hemodynamic performance and vascular compliance. Due to the low melting point of PCL, some methods of common paraffin embedding and graft sterilization are not suitable for this polymer and low-heat histoprocessing and sterilization techniques should be preferred. High-temperature tissue processing is not suitable for evaluating the graft fiber degradation rate and scaffold architecture, but it is appropriate for histological analysis, cellular penetration and ingrowth description.

## Figures and Tables

**Figure 1 polymers-14-03313-f001:**
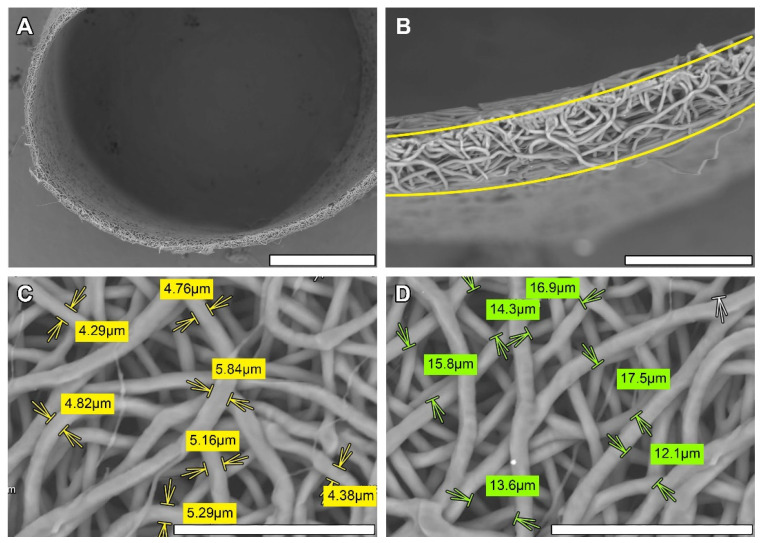
SEM images of monolayered electrospun PCL graft: cross section, bar—500 µm (**A**); graft wall, bar—100 µm (**B**); inner surface before implantation with fiber diameter (**C**), yellow arrows mark fiber margins; and pore size (**D**), green arrows indicate pore margin measurements, bar—50 µm.

**Figure 2 polymers-14-03313-f002:**
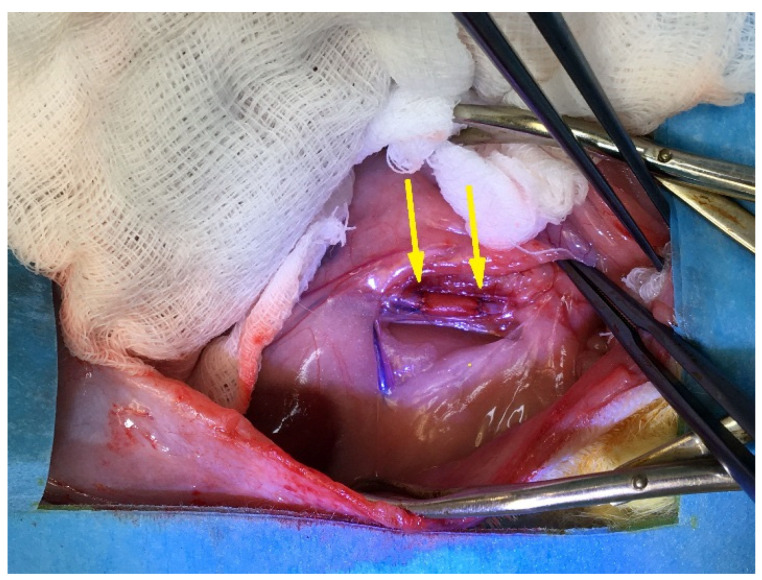
Implantation in rat abdominal aorta. Yellow arrows indicate margins of the PCL graft.

**Figure 3 polymers-14-03313-f003:**
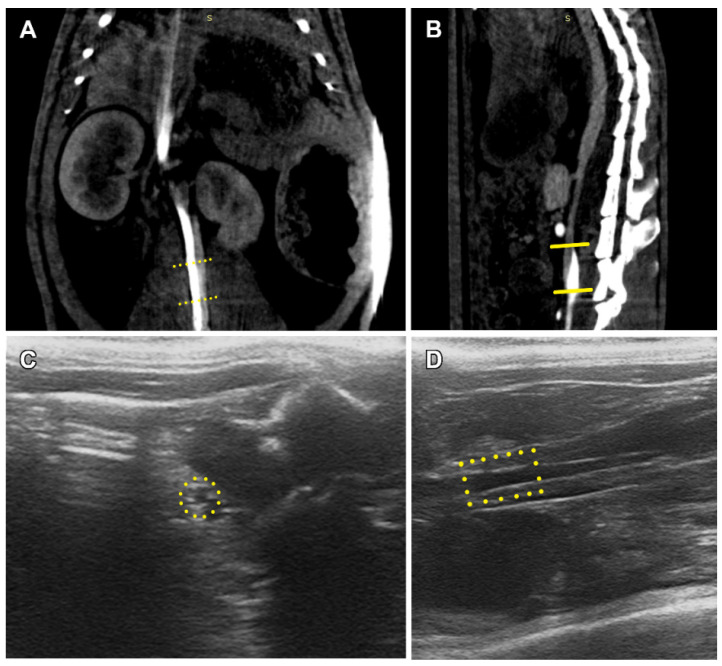
Functional diagnostic images of implanted graft, 90 days after implantation. CT, coronal section (**A**); CT, sagittal section (**B**); ultrasonic image, cross section (**C**); ultrasonic image, sagittal section (**D**). Yellow frames mark graft position.

**Figure 4 polymers-14-03313-f004:**
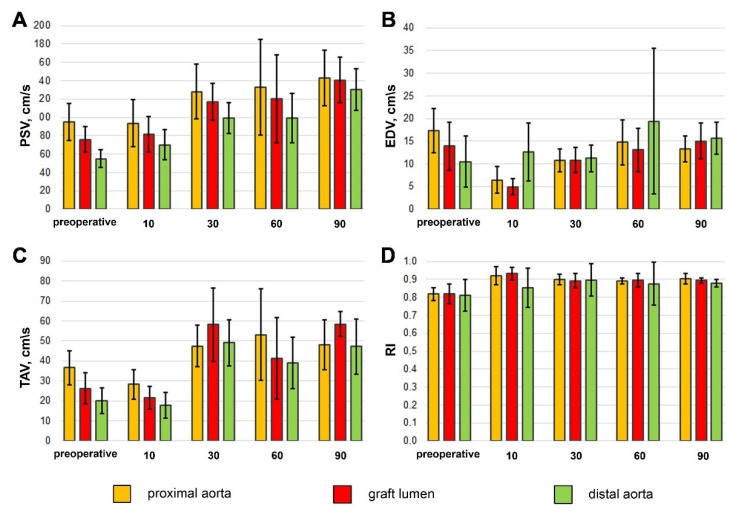
PSV (**A**), EDV (**B**), TAV (**C**) and RI (**D**) changes by follow-up time. The results are given as mean (M) and standard deviation (σ).

**Figure 5 polymers-14-03313-f005:**
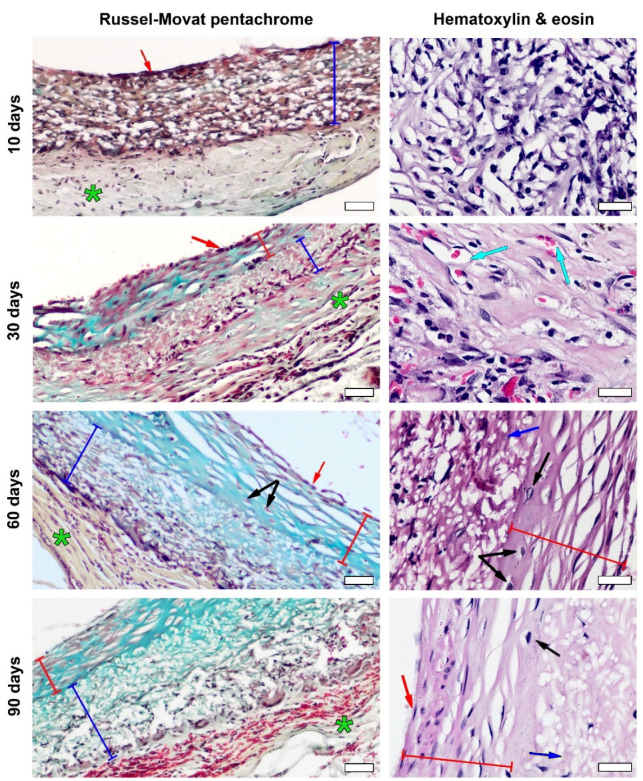
Histological picture of samples by follow-up period. Russell–Movat pentachrome staining, magnification—×100, bar—50 µm, H&E staining, magnification—×400, bar—20 µm. The red arrows indicate luminal surface, red lines—intimal hyperplasia, dark blue lines—PCL graft, green stars—adventitia, turquoise arrows—neoangiogenesis, black arrows—chondroid metaplasia.

**Figure 6 polymers-14-03313-f006:**
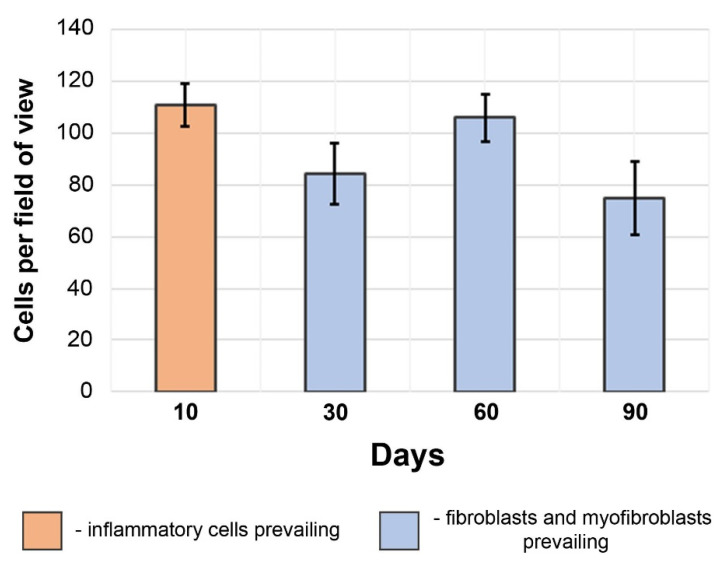
Scaffold cellularity dynamics. The predominant cell types are different: inflammatory cells (10 days) or fibroblasts and myofibroblasts (30–90 days). The results are given as mean and standard deviation (σ).

**Figure 7 polymers-14-03313-f007:**
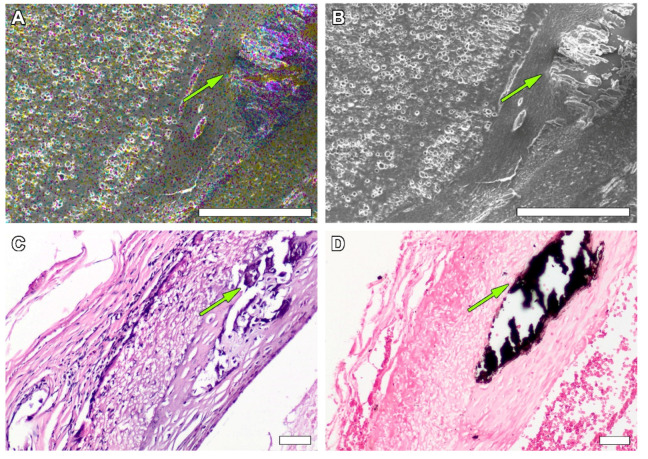
SEM element analysis of the calcification that developed under the layers of endothelial tissue and intimal hyperplasia at 60 days after implantation. The blue points indicate phosphorus, yellow—oxygen, bright pink—calcium, bar—250 µm (**A**). Overall SEM image of crystal clusters, bar—250 µm (**B**). H&E image of mineralization point, magnification—×100, bar—50 µm (**C**). von Kossa staining of mineralized area, black staining indicates presence of calcium, magnification—×100, bar—50 µm (**D**). Green arrows indicate crystal clusters.

**Figure 8 polymers-14-03313-f008:**
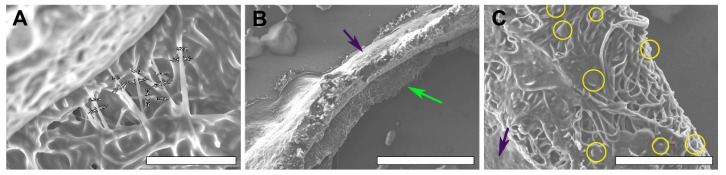
SEM image of a dried graft section at 90 days after implantation. Fiber thinning, bar—50 µm (**A**); graft wall, bar—500 µm (**B**); cellular infiltration, bar 100—µm (**C**). The dark purple arrows are luminal surface, green arrow is adventitial surface, yellow circles enclose migrating cells.

**Figure 9 polymers-14-03313-f009:**
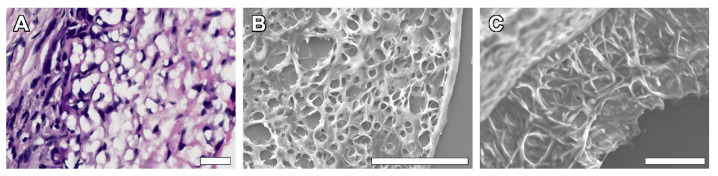
Comparative images of the samples explanted at 90 days after implantation. Paraffinized H&E stained section, light microscopy, magnification—×400 (**A**), SEM images of deparaffinized section (**B**) and dried section of the same sample, magnification—×600 (**C**).

## Data Availability

The data presented in this study are available within the article.

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
