# Peer review of "In Vivo Evaluation of PCL Vascular Grafts Implanted in Rat Abdominal Aorta"

_polymers, 2022, doi:10.3390/polym14163313_

Round 1

Reviewer 1 Report

Summary: The goal of this project was to evaluate the in vivo performance and fluid mechanics parameters for electrospun PCL scaffolds implanted as an aorta substitute. This work can be used to illustrate how the native biological system adapts to implanted materials, remodels implanted materials and how these implanted materials survive within biological systems. The group also illustrated that these scaffolds can be implanted within rat models for 90 days and show successful outcomes. However, the work is largely descriptive and there are missed opportunities to quantify data presented in favor of qualitative data. Additionally, these experiments do not appear to be controlled properly as there was no discussion of sham surgery animals and/or animals that did not undergo any surgical procedures. It is also possible that the group uses other approved grafting material to check performance against their PCL scaffolds. The lack of this ‘control’ data lowers the overall impact of the presented research as there is little ability to understand the effects of PCL graft implant. Additionally, there are other concerns which will be described below.

1. The use of the phrase ‘short-term’ is probably not needed in this case. Compared with many other experiments 90-days, would be considered very long-term. The authors may want to delete the phrase short-term from the title and clarify its usage throughout the document or identify a more accurate term that can be used throughout the document.

2. The abstract has a number of concerns. First, there is very limited discussion on the work that this group undertook. In parallel to this, this reviewer could not find a discussion of any key results or the impact of these key results. Additionally, there is little connectivity between the sentence structure, which makes reading the abstract very awkward. Finally, this reviewer would suggest moving away from using the phrase gold standard, in relation to electrospun scaffolds. This is debatable at best but relatively inaccurate based on many reports that illustrate that electrospun scaffolds do not withstand repeatable mechanical stimuli well.

3. This reviewer would also suggest rewriting the introduction. There are many unsubstantiated claims, combined with poor word choices, which make the reading of the introduction very challenging. For instance, in the first sentence the phrase ‘most perspective direction’ does not make sense and would be very controversial. The second sentence does not follow logically from the first. Same with the third sentence following the second … This reviewer would highly suggest that the introduction be looked at and refocused more closely on the study that is being presented here.

4. Also in the introduction, this reviewer would refrain from saying that a single particular study was the most interesting. It is speculation, debatable and does not add anything to the document.

5. The final sentence of the introduction seems to contradict the abstract and the data presented; since it only focuses on 10/90 days and ignores the intermediate points that were investigated. Additionally, this reviewer was not sure how one would define ‘complex’ hemodynamic and structural properties and whether or not what this group investigated was complex properties or not.

6. Section 2.5 was very unclear to me and it was very challenging to follow the dosing and administration of medication to the animals. This reviewer would suggest editing this section for clarity so the work can be judged and duplicated more easily. 

7. Similar in section 2.5 (or it can be the first results described), this reviewer would suggest the inclusion of images illustrating the surgery. Therefore, successful implantation of the graft can be viewed by the readers of the manuscript.

8. For Figure 1: The thickness of the graft shown in panel B does not match the average thickness reported and appears to be quite smaller than 54um. This reviewer would suggest showing a scaffold that is closer to the average thickness. Also, the text in images C and D is not legible under a normal zoom ratio. This reviewer would suggest removing some of these values and making only a few of these larger to illustrate the measurements. In parallel, it may be possible to combine Figure C/D as they appear to be the same scaffold location, just with fiber diameter measurements and pore size measurements identified … there is no reason that these need to be two separate images (perhaps the different values can be color coded on the single image). Additionally, this reviewer could not find sufficient information on how pore size was approximated. As these scaffold images are three dimensional substrates projected onto two dimensions, and pores are effectively two dimensional, the calculations need to be sure to exclude fibers that cross the pores in the uppermost plane. How was this handled numerically. Also, it seems that we only have one dimension recorded for the pore; was this single dimension assumed to be the side of a square, the diameter of a circle, etc.  Further, when zooming in to look at the pore definitions, there are clearly pore recordings that are inaccurate by taking into account distance between fibers in different planes. This reviewer would highly suggest that: 1) more information is provided regarding pore size calculations and 2) that the group verifies that the pore calculations were done accurately as this is not apparent now. Not only is the data that is shown in panel D unclear, it is also clear that at least 50% of the dimensions are not accurate for pore calculations.

9. Figure 3, please describe what the different colors represent. Were statistical analyses conducted over time (e.g. for PSV, it visually appears that differences may exist between 10 days and 90 days).

10. Can any of the data discussed within Figure 4 be quantified. For instance the thickness of the various layers over time would provide information about remodeling, endothelium growth, degradation of PCL, etc.

11. This reviewer also did not understand what is being shown in Figure 5. What are the units on the y-axis, how could one describe a decrease in cellularity, followed by an increase, followed by a decrease again? In practice, when data points are joined directly (e.g. the blue line on this curve), this means that the group is confident that this saw tooth pattern is what is expected. Wouldn’t you want to represent this data with a linear trend line that represents the general trend over time. This figure is also very challenging to interpret without the controls that were mentioned above (e.g. sham surgery, no surgery, other graft material …)

12. In the discussion, the statement, a thickness of 54.1 +/- 7.374 um was chosen as optimal seems odd. What were the parameters that determined this and how can one state that a scaffold with an average thickness of 54.1 (and given standard deviation) is optimal; this suggests that a scaffold that had the same mean thickness but a different standard deviation would not be optimal; can this finding be substantiated? A few sentences later, there is a citation to abdominal thickness of 50.4+/- 3.3. What are the units and what is the species that this was found in?

13. Figure 8 appears out of place in the discussion and it is unclear to this reviewer what it adds to the discussion that was presented as the goals of this work. This reviewer would be in favor of removing this entirely but at a minimum moving it out of the discussion section.

14. This review would also suggest editing the discussion/conclusion for clarity. There are some confusing and controversial statements. 

Reviewer 2 Report

This study demonstrated some limitations of PCL vascular prostheses prepared by electrospun including slow biodegradation, mineralization risks and ineffective cellularization. I believe this is an interesting work to readers who work in vascular graft. Unfortunately, some evidence shown here are not well characterized, explained and discussed. Please answer and modify some major and minor points shown below before further consideration.

Major:

1.     P1, the references (ex: rf16 (2008), rf17(2012)) are too old. Please cite and review literature published within 5 years to have more complete review on this area, especially in PCL prepared and sterilized by various methods.

2.     P8, this paragraph “The peak of remodeling process occurred on the 30th day of observation……. (Fig.4)” describes the inflammatory response increased and migration of giant cells indued by PCL graft. Pls stain specific antibodies to visualize the interaction between specific kinds of immune cells and PCL.

3.     “The most stable scaffold ingrowth with high cellularity was registered at 60 days after implantation (Fig. 4, 5).” “At the same time, at the 60th day of observation first signs of prosthesis calcification were found on the H&E slides”. The stining images did not provide sufficient evident to quantitatively demonstrate the cellularity, calcification or chondroid metaplasia. Pls use specific antibodies to address this interesting observation.

4.     In conclusion part, authors mentioned “Due to physical properties of PCL, some methods of common laboratory analysis and graft sterilization are not suitable for this polymer and low-heat sterilization techniques should be preferred. “Could this PCL material be unsuitable for vascular graft due to this low-heat sterilization methods used in this study? Have authors tried to change material properties of PCL? What are the results? 

Minor:

1.     P1, missing words: TEVGs since the end of “XX” century. 

2.     P6, Fig3, what are the bars with different colors?

Reviewer 3 Report

Polymeric graft biocompatibility and safety profile defines grafting success. Lack of cellularisation and unfavourable biodegradation profiles of vascular scaffolds can trigger inflammatory processes resulting in excessive hyperplasia and occlusion, fibrotic processes as well as calcification.  Authors investigated PCL-based vascular scaffold cellularisation and mechanical properties post-engraftment   in the in vivo rat model. Importantly, they zoomed at the changes occurring shortly after engraftment (10-90 days) and observed rapid cell accumulation in the vascular regions around the scaffold peaking at day 60 followed by the cellular loss by day 90. Importantly, no changes in mechanical properties were observed by day 90 (with N=40) indicating that the scaffold parameters fits the required functionality. Interestingly, they defined the spatial localisation of the degenerative processes by showing initiation of calcification in the luminal part and active phagocytic cells accumulation at the adventitial side. These insights are important to design next generation of vascular scaffolds with the reduced propensity for hyperplasia, inflammation, calcification and fibrosis.

However, an additional clarification of cellularity index is needed to understand the level of cellularisation process. Also, key data/conclusion should be included in the abstract. Minor changes can be considered across the manuscript.  

Major

1. Abstract summarise the rational for the study but key deta/conclusions are not included.

2. Figure 5 shows cellularisation processes. Is it correct that the “Cellularity” number is a number of counted cells? Is there any way to normalise these data (eg per graft or area?). Also, is there a statistical difference between day 30, 60 and 90? Figure 4 day 90 indicates quite a large number of accumulated cells but Figure 5 shows cellular loss. How it can be explained? Finally, what are the cells in large quantities at day 10 as figure 4 shows little evidence of infiltration?  

3. Authors  showed the thinning of polymeric fibers after 90 days – is it statistically significant as compared to Fig 1 data?

Minor

1. Page 1. Introduction “This process should…” Please clarify what process?

2. Page 1. Introduction “All of the above…” May be consider to replace with a short summary of the key polymeric graft’s properties?

3. Page 2. Introduction “grafts at 10 and 90 days” – other times points can be included here?

4. Figure 4 shows the formation of neointima as well as adventitia over the time. Is it possible to quantify these processes to see if there is a difference between day30 and day90?

5. Figure 6 has 3 panels but only 2 (A and B) are described in the figure legend.

6. One of the figures (Fig 8) was included in the discussion – is there any opportunity to move it to “Results” section?

7. Page 11, Discussion Type “..specimens such as bon…” bone?

Reviewer 4 Report

The manuscript “In vivo Evaluation of PCL Vascular Grafts Implanted in Rat Abdominal Aorta” authored by Dokuchaeva et al. represents an interesting work concerning the development of vascular grafts for abdominal aorta applications. The authors performed in vivo studies in rat models and they were to verify that the construct was able to induce the endothelialization of cells on it and did not show any sign of rupture or alteration up to 90 days. The work seems interesting but before being considered for publication, the authors should answer to the following questions.

-       In the introduction part, what does “three ply-vessel” mean?

-       “Yet, for successful performance of such medical device remodeling process should be completed within 1 year, otherwise the prosthesis will undergo fibrosis and calcification”. This sentence is not well written and what does “should be completed within 1 year? Some applications do not require a degradable medical device, or some other applications might use medical device for short or long term application. Moreover, what is relation between 1 year and fibrosis??? Please explain.

-       In the paragraph about “in vivo implantation”, did the authors perform tests and analysis to ensure that the abdomen aorta is healthy and intact before performing the surgery? Please discuss.

-       The authors are invited to remove “Histo-“ and keep “cytotoxicity”.

-       In paragraph 2.2. What were the environmental parameters in terms of temperature and relative humidity?

-       Why did the authors not perform mechanical tests on the fabricated scaffold?

-       The authors are invited to change the following sentence from “0,05 mg/kg of atropine” to “Atropine sulfate (0.05 mg/Kg) was administrated.

-       Fig.S1 must be improved in terms of quality and content. The authors should provide representative figures concerning the experimental and operational surgery. The authors additionally are invited to add macroscopical picture of the fabricated graft.

-       What about the scaffold properties in terms of ultrastructure and mechanical properties after the sterilization process? Please discuss and add more information concerning this issue.

-       Why did the authors not consider a Control group to compare the obtained results with healthy rats?

-       In figure 4, what does the blue line represent? Please discuss.

-       “The thickness of the PCL matrix was infiltrated with lymphocytes, macrophages and giant cells; very few fibroblasts could be found. The pale yellow staining marked the presence of collagen in the forming adventitia.” From the provided pictures shown in Figure 4, how could the authors say that the infiltrated cells were lymphocytes and macrophages? Did the authors perform immunohistochemical analysis using immune cells related markers? Please discuss.

-       The provided pictures in Figure 4 are not clear to be understood. The authors are invited to add new clear pictures with improved resolution.

-       p-value” should be written in italic. Please adjust overall the text.

-       The authors should have been performed more analysis concerning ECM deposition to see the effectiveness of the vascular grafts in inducing cell differentiation and ECM deposition. Please discuss.

-       In the discussion part the authors talked about the difficulties they faced during sterilization at high temperature. Considering that this Journal “polymers” is related to material fabrication and characterization together with their applications, it would be important that the authors showed the discussed results concerning the effect of sterilization temperature on PCL integrity. Please discuss.

Round 2

Reviewer 1 Report

            This reviewer appreciates the effort made by the group to address some of the critical limitations of this study; however, significant concerns still remain as the group did not address the major scientific weaknesses in this work. The following concerns remain:

1)    The description of pore size calculation with the additional figure does not help the reader understand how this data was calculated. Pore size is not the average of the distance between sides of a polygon formed and it can be calculated directly from images once pore boundaries are identified. This is fairly standard and many software exist that can do this automatically. Furthermore, the addition of panel D to Figure 1, while helpful visually, again does not illustrate whether or not pore size is being calculated properly. All this panel illustrates is that the group can measure the distance between two points on a microscopy image; again fairly standard and almost every imaging software should allow this to happen. Without an actual description and illustration of how pore size was calculated, the reader is left without any clear picture of what this group is reporting.

2)    This review appreciates the addition of the color coding to Figure 4, however, it still remains unclear, 1) why there were no statistical analyses undertaken, 2) why the values of PSV and EDV steadily increased over time, 3) what are the pre-surgery values for PSV, EDV and RI, 4) the absence of a surgical control (or any other control) is immensely problematic as the reader has no idea what the baseline and standard values are for these types of recordings. Without these values the relevance of this work is going to always be questionable. Further, statistical analysis of the data should be a requirement as it is immensely challenging to draw any conclusion without statistical support.

3)    Further, with increasing cardiovascular velocities, we would expect more damage to the endothelial layer, unless the values are increasing to their normal pre-surgical values (again, as discussed in point 2, without this data the reader has no way to gauge the significance of this increase). If it is true that the velocity values are increasing in an abnormal way, this would explain the increase in inflammatory cells that were observed. Further, if this increase in velocity and inflammation continues to increase, it is likely that the device will fail.

4)    There appears to be a contradiction within the writing; Figure 5 shows hyperplasia, but the text says that no hyperplasia was found (perhaps in reference to day 10). As pointed out above, if there was no hyperplasia at an earlier time point, and then hyperplasia was found at a later time point, this would be a significant cause for concern.

5)    There also appears to be a contradiction within the writing associated with Figure 6; it is claimed that the most stable scaffold ingrowth (which is not defined) and cellularity is found on day 60. The only data we have access to, is the cellularity which is highest at day 10. Further, as mentioned in the previous review, these data points cannot be connected directly by a trend line unless the authors are confident that these are the exact numerical values for cellularity at days 10, 30, 60 and 90 and that the changes in cellularity will follow the lines shown. Since this is not likely to be true, the authors should represent the data with a trend line and justify why they used that particular trend line (e.g. linear vs. exponential).

6)    Also, figure 6 is very challenging to interpret as there are no controls, other scaffolds or pre-operative measurements to compare this data to.

Reviewer 2 Report

Authors did properly answer all question and significantly improve the quality of this paper
